# Metabolic Reprogramming Induced by Aging Modifies the Tumor Microenvironment

**DOI:** 10.3390/cells13201721

**Published:** 2024-10-17

**Authors:** Xingyu Chen, Zihan Wang, Bo Zhu, Min Deng, Jiayue Qiu, Yunwen Feng, Ning Ding, Chen Huang

**Affiliations:** 1Dr. Neher’s Biophysics Laboratory for Innovative Drug Discovery, State Kay Laboratory of Quality Research in Chinese Medicine & Faculty of Chinese Medicine, Macau University of Science and Technology, Taipa, Macao SAR 999078, China; xiaozeshiwoxiaodi@outlook.com (X.C.); mingyangzhan@outlook.com (Z.W.); zhubo6824@sina.com (B.Z.); 3230006748@student.must.edu.mo (J.Q.); morphy1678@outlook.com (Y.F.); 2009853zc211002@student.must.edu.mo (N.D.); 2Faculty of Health Sciences, University of Macau, Taipa, Macau SAR 999078, China; mindeng@um.edu.mo

**Keywords:** metabolic reprogramming, aging, metabolic plasticity, tumor immune microenvironment, pan-cancer, glioma, scRNA sequencing analysis

## Abstract

Aging is an important risk factor for tumorigenesis. Metabolic reprogramming is a hallmark of both aging and tumor initiation. However, the manner in which the crosstalk between aging and metabolic reprogramming affects the tumor microenvironment (TME) to promote tumorigenesis was poorly explored. We utilized a computational approach proposed by our previous work, MMP^3^C (Modeling Metabolic Plasticity by Pathway Pairwise Comparison), to characterize aging-related metabolic plasticity events using pan-cancer bulk RNA-seq data. Our analysis revealed a high degree of metabolically organized heterogeneity across 17 aging-related cancer types. In particular, a higher degree of several energy generation pathways, i.e., glycolysis and impaired oxidative phosphorylation, was observed in older patients. Similar phenomena were also found via single-cell RNA-seq analysis. Furthermore, those energy generation pathways were found to be weakened in activated T cells and macrophages, whereas they increased in exhausted T cells, immunosuppressive macrophages, and Tregs in older patients. It was suggested that aging-induced metabolic switches alter glucose utilization, thereby influencing immune function and resulting in the remodeling of the TME. This work offers new insights into the associations between tumor metabolism and the TME mediated by aging, linking with novel strategies for cancer therapy.

## 1. Introduction

Aging is a strong risk factor for tumorigenesis, with more than 60% of cancers occurring attributed to patients aged 60 and over [1]. One interesting example is that Werner syndrome, a premature aging syndrome occurring in adults, had an increasing incidence of cancer [2]. Accumulating evidence indicates that metabolic reprogramming is a hallmark of both aging and tumor development [3,4,5]. Some evidence also indicates that the ability of neurons in the aging brain to utilize glucose decreases [6]. The Warburg effect, a well-known metabolic reprogramming event, takes up glucose through glycolysis rather than oxidized phosphorylation (OXPHOS), even in oxygen-rich environments, to maximize biomass and rapidly provide energy for tumor cell growth and proliferation [7,8]. Furthermore, recent reports suggest that aging-driven metabolic reprogramming in different cell types significantly affects cellular functions, thus providing perfect environments for premalignant cells [9]. In particular, the dysfunction of one-carbon metabolism led to the adequate priming and effector differentiation of pre-effector-like T cells, thereby exhibiting resistance to anti-PD-1 therapy [10,11]. However, the comprehensive investigation into the association between aging-related metabolic alterations and the tumor microenvironment (TME) remains poorly explored. Focusing solely on the increase or decrease in the activity of a single pathway may overlook the crosstalk between metabolic pathways (MPs), and this dynamic and plastic interaction between MPs influences the ecology of the TME, resulting in a unique transcriptional landscape of the tumor. Herein, we aim to dissect the heterogeneity of tumor tissues associated with aging and explore how specific aging-related metabolic shifts in particular cell types contribute to the development of cancers, which may offer promising avenues for cancer treatment.

Briefly, we firstly comprehensively depicted the aging-related metabolic plasticity across 17 different cancer types using an in silico framework developed by our previous study, MMP^3^C (Modeling Metabolic Plasticity by Pathway Pairwise Comparison) [12,13,14]. MMP^3^C analysis facilitates us to characterize aging-related metabolic plasticity events associated with tumor initiation and progression, thereby laying the foundation for understanding the significance of metabolic reprogramming in the context of aging and cancer. The MMP^3^C framework initially calculated the MP activity score based on gene expression profiles and the built-in pathway protein-to-protein interaction (PPI) network weight value. Then, an intra-sample comparison of two MPs was conducted to eliminate the batch effect of samples. Subsequently, significant metabolic switches were identified based on chi-square tests among different aging stages. The tendency of metabolic switches between two MPs was evaluated using odd ratio values among different aging stages. Additionally, we explored metabolic reprogramming within the TME at single-cell resolution for tumors highly associated with aging, aiming to provide promising avenues for cancer treatment.

## 2. Materials and Methods

### 2.1. Datasets

The pan-cancer gene expression, methylation, and clinic information datasets produced using the HiSeq Illumina platform, including 26 solid cancer types, were downloaded from the UCSC Xena Database (https://xenabrowser.net/datapages/, accessed on 26 April 2023). Tumor type-specific age quartiles were considered age group boundaries for the cancer genome atlas (TCGA) cohorts, defining the younger group and the older group by the lower and upper quartiles, respectively.

The single-cell RNA-seq data (GSE163120) with seven samples of primary glioma were downloaded from the gene expression omnibus (GEO) database (https://www.ncbi.nlm.nih.gov/geo/query/acc.cgi, accessed on 26 April 2023) from a previous study [15]. Additionally, cptac-3 is available at DATE from https://registry.opendata.aws/cptac-3 (accessed on 1 May 2024). The age group limits of single-cell samples were set by fixed thresholds and were consistent with samples from bulk sequencing.

### 2.2. Immune Infiltration Analysis by ssGSEA

The 28 immune cell infiltration distribution was estimated by the immune cell signatures [16] using the single sample gene set enrichment analysis (ssGSEA) method. The Gaussian kernel was employed with TPM data.

### 2.3. DNAm Age Calculation

The epigenetic age of aging-related cancer was computed via the ‘methylclock’ R package [17]. This tool infers epigenetic age based on 353 CpG sites by the Horvath method [18]. In addition, predicted ages with a correlation lower than 0.8 with the internal gold standard were discarded from downstream analysis.

### 2.4. RNA Age Calculation

The FPKM RNA-seq datasets were obtained from the UCSC Xena database to determine transcriptional age using the ‘RNAAgeCalc’ R package [19]. The regression model for transcriptional age utilized the Dev signature, which encompassed coefficients for genes that had the largest variance across samples.

### 2.5. Trajectory Analysis Based on Bulk Sequencing

The pan-cancer cohort was leveraged to infer the tumor development with the age covariate via the ‘PhenoPath’ package [20]. Then, principal component analysis (PCA) was processed via the ‘stats’ [21] R package to visualize the trajectory of tumor development with aging.

### 2.6. Single-Cell RNA-Seq Analysis

#### 2.6.1. Data Preprocessing, Cell Clustering, and Annotation

Utilizing the ‘Seurat’ [22] R package (V 5.1.0), we preprocessed single-cell gene expression data. The VlnPlot() function was employed to visualize three quality control indices and filter low-quality cells by cells with fewer than 200 or more than 20,000 UMIs, gene numbers above 4500, mitochondrial gene percentages exceeding 10%, and the gene-to-UMI ratio below 0.8. The doublets were also identified and excluded via the ‘scDblFinder’ R package [23]. After the quality control process, the gene expression matrix underwent library size normalization via the NormalizeData() function with the “LogNormalize” method and a size factor of 1000. Highly variable genes were identified using the “vst” method in FindVariableFeatures(), and principal component analysis (PCA) was performed with the top 2000. The optimal number of PCs was determined by PCElbowPlot(). Cell clustering was executed with FindNeighbors() and FindClusters(), followed by dimensionality reduction using RunUMAP() and RunTSNE() for visualization. DEGs were identified within each cluster using the Wilcoxon rank-sum test in FindAllMarkers() or FindMarkers(), with a *p*-value threshold of 0.05. These DEGs were then used to annotate cell subtypes, referencing the CellMarker V2.0 database and prior research [15].

#### 2.6.2. Integration of Multiple Single-Cell Transcriptome Data Cohorts across Samples

The IntegrateLayers() function in the ‘Seurat’ R package was utilized to integrate multiple single-cell transcriptome data cohorts from the GEO and cptac3 database. Different parameters consisting of ‘harmony’, ‘CCA’, ‘RPCA’, and ‘scvi’ were performed, and thus, various integrated methods were compared to find the optimal method for the batch effect of datasets.

#### 2.6.3. Trajectory Analysis to Infer T Cell Fates in the Aging Process

The trajectory inference analysis using the ‘Monocle2′ [24] R package (V2.3.6) revealed the different T cell fates in the aging process. T cell clustering and annotation were further performed for the T cells. DEGs among T cell subtypes identified by Seurat were regarded as the ordering genes. We established the root state as naïve T cells and applied the orderCell() function from Monocle2.

#### 2.6.4. Cell Type Enrichment of Various Age-Based Subgroups

To quantify the enrichment of immune cell types among different age-based subgroups, we compared the observed and expected cell numbers in each age-based subgroup by calculating the R_o/e_ (the ratio between observed and expected cell numbers) value via the ‘epitools’ R package (https://cran.r-project.org/web/packages/epitools/index.html, accessed on 23 May 2024).

#### 2.6.5. Detecting Malignant Cell Based on Genomic Copy Number Inferring

To detect the malignant cells in single-cell RNAseq data, genomic copy number inferring was performed using the ‘copykat’ R package (V1.1.0). The ‘n.cores’ parameter was configured to 20 for parallel computation, and other parameters were set to default values. Due to memory limitation, cells were divided into various batches for inferring copy number.

### 2.7. Proteomics Analysis

We calculated the MMP^3^C scores for the glioma (GBM) proteomics dataset (downloaded from https://pdc.cancer.gov/pdc/browse, accessed on 10 October 2024) based on MPs (genes within the pathway expressing at least 30%) and conducted the chi-square test or Fisher’s exact test among aging-related subgroups to screen for metabolic switches with *p*-values < 0.05. These metabolic switches were then plotted in a Venn diagram alongside those from the GBM RNA-Seq data in the TCGA database (Appendix A). Additionally, Pearson’s correlation coefficients were calculated for the MMP^3^C scores of the GBM proteomics dataset with respect to age (Appendix A).

### 2.8. Statistical Analysis

Statistical analyses were performed using R software, specifically version 4.3.1. To assess differences between categorical variables, either the chi-square test or Fisher’s exact test was employed. Significance was determined at the two-sided *p*-value threshold of 0.05, which was considered statistically meaningful.

## 3. Results

### 3.1. Aging-Related Metabolic Alterations: Insights from Pan-Cancer Bulk Sequencing Transcriptome Analysis

Initially, the univariate Cox regression analysis was utilized to examine the correlation between aging and survival outcomes (time and status) across 26 solid cancer types from the TCGA database [25]. To delineate the effect of aging, covariates such as sex, race, tumor stage, etc. were included in the Cox survival model. The results showed that 17 cancer types were correlated to age (adjusted *p*-value < 0.05) after eliminating the influence of covariates. Among them, GBM with significant *p*-value (2.970067 × 10^−15^ and low-grade glioma (LGG, 3.329830 × 10^−12^; Figure 1A) were identified, which was consistent with many previous studies [26]. Then, these aging-related cancers were utilized to investigate aging-related metabolic plasticity events by comparing adult and aged patients using the MMP^3^C framework (Figure 1B,C). Briefly, the 3486 candidate MP pairs (yielded by pairwise pairing between 84 Kyoto Encyclopedia of Genes and Genomes (KEGG) MPs; Appendix A) were subjected to MMP^3^C toward global screening of metabolic plasticity events. The results displayed that distinct cancer types exhibited a diverse number of significant metabolic plasticity events (Figure 1B), implying an organized heterogeneity of metabolic plasticity across cancers. Nonetheless, many significant metabolic plasticity events were identified in several specific cancers simultaneously (Figure 1B,C). For instance, a metabolic plasticity event composed of propanoate metabolism and fatty acid degradation was consistently observed in head and neck squamous cell carcinoma (HNSC), LGG, lung squamous cell carcinoma (LUSC), stomach adenocarcinoma (STAD), uterine corpus endometrial carcinoma (UCEC), and thyroid cancer (THCA), whereas this metabolic plasticity event was reversed in kidney renal clear cell carcinoma (KIRC) and thyroid cancer (THCA; Figure 1B,C). In addition, steroid hormone biosynthesis prevalently increased flux in multiple cancer types. In GBM and LGG, the flux of fatty acid elongation and biosynthesis decreased in the older patients but increased in the younger patients. This might suggest that older patients had low immunity in the glioma since lipid biosynthesis was required for the activation of T cells [27]. Thereafter, to further identify the consensus of dysregulated MPs in the pan-cancer cohorts, the top 200 significant metabo-plastic events for each cancer type ordered by the adjusted *p*-value yielded by MMP^3^C were selected to construct a metabo-plastic network (Figure 1D; Appendix A). Subsequent network analysis via Gephi [28] disclosed many critically altered MPs (hub nodes in the network; Figure 1D) in the process of senescence, which involved the steroid hormone biosynthesis, oxidative phosphorylation, galactose metabolism, pantothenate and CoA biosynthesis, drug metabolism, and glycosylphosphatidylinositol (GPI)-anchor biosynthesis, implying their importance to senescence (Figure 1D). In addition, some of the altered MPs were validated by the comparison of the single MP activity score computed by MMP^3^C between old and young groups via the ‘limma’ [29] R package (Figure 1E and Appendix A), i.e., glucose metabolism, energy metabolic pathways, etc., disclosing high confidence of MMP^3^C analysis.

### 3.2. The Metabolic Switch Correlates to Molecular Features of Senescence in Aging-Related Cancers

To examine the correlation between the metabolic switch and molecular aging, as well as senescence, initially, the methylation age [17] and RNA age [19] were assessed using the ‘methylclock’ and ‘RNAAgeCalc’ R packages, respectively. Then, the Pearson correlation analysis was performed. The results showed that many metabolic switches exhibited a significant correlation with methylation age or RNA age (Appendix A), i.e., Nitrogen metabolism increased in contrast to Glutathione metabolism, which decreased with the methylation age (R = 0.45, *p*-value < 0.01; Appendix A), and fatty acid biosynthesis declined in contrast to fatty acid degradation, which increased with the RNA age (R = −0.47, *p* value < 0.01; Appendix A). Furthermore, on the basis of the MP pair profile, pseudo-temporal analysis was conducted (chronological age was set as a covariate) via the ‘Phenopath’ R package [20]. As shown in Figure 2A, we identified distinct patterns of the evolving tumor metabolism among different age-based groups (the patients were categorized into three age-based groups, young, middle, and old, based on the lower and upper quartiles of their age distribution) [26]. Subsequently, correlation analysis between MP pair activities and pseudo-temporal scores disclosed several metabolic plasticity events associated with the changes in root time (Figure 2B). For example, Taurine and hypotaurine metabolism greatly decreased in the older group, while various types of N-glycan biosynthesis increased compared to the younger group, exhibiting opposing directions of metabolic transformation. Notably, distinct immunological activity among three age-based groups was detected through immune infiltration analysis based on the ssGSEA method. The infiltration scores increased with aging progression in most immune cell categories, except for Eosinophil and Monocyte (Figure 2C); in particular, the increased infiltration of effector memory CD4+ T cell was found in the young group compared with the remaining two groups (Figure 2C). This result implies a strong correlation between aging and immunity in patients with tumors.

### 3.3. The Landscape of Tumor Microenvironment Showed Distinct Discrepancy among Aged-Based Subgroups

The above analysis indicated that aging exhibited a high degree of association with metabolism and immunity in cancer, and we thereby seek to explore how the interplay of immune and metabolism reprogramming during aging impacts tumorigenesis using integrated scRNA-seq data of GBM, which contain 24 primary tumor samples with different chronological ages. The reason that we chose GBM is that GBM has been demonstrated to exhibit a strong correlation with senescence [30]. Also, our previous Cox regression analysis demonstrated a significant correlation between GBM and aging (Figure 1A), which is corroborated by proteomic findings and transcription-based metabolic shift characteristics (Appendix A). Conventionally, all GBM patients were divided into three aging-related groups: young (age < 50, n = 6), middle (50 ≤ age < 65, n = 12), and aged (age ≥ 65, n = 6; Appendix A). After filtering low-quality cells, we established the scRNA-seq atlas consisting of 181,286 qualified cells. The uniform manifold approximation and projection (UMAP) clustering mainly generated nine cell types, including T cells (*TRBC2*, *CD3E*, and *CD3G*), oligodendrocytes (*PLP1*, *MBP*, and *UGT8*), neuroendocrine cells (*CHGA*, *NRSN1*, *NDRG4*), macrophages (*C1QA*, *C1QB*, and *CSF1R*), Fibroblasts (*COL1A2* and *COL3A1)*, Astrocytes (*AQP4*, *CLU*, and *CHI3L1*), Oligodendrocyte precursor cells (OPCs; *ALDH1L1*, *BCAN*, and *MEGF11*), Neurons (*TOP2A*, *RRM2*, *STMN2*, and *DLX6-AS1*) and B cells (*MS4A1*, *MZB1*, and *IGKC*; Figure 3A,B, Appendix A). Younger groups showed an increased presence of the stromal cells (neuroendocrine cells and neurons) and OPCs (Figure 3C). Elder groups had a higher abundance of oligodendrocytes and exhibited increasing infiltration of T cells and macrophages in the TME (Figure 3C), implying a close correlation of T cells or macrophages with aging. This result is in line with our previous ssGSEA analysis using bulk RNA-seq data (Figure 2C). Therefore, different kinds of immune cells (T cells and Macrophages) were further curated for cell re-clustering and annotation based on their universal marker gene [15,31,32] and CellMarker2.0 database [33].

The UMAP clustering of all 12,144 T cells generated 13 T cell subclusters that were identified, and nine main cell types were annotated, consisting of central memory T cells (Tcm), effector T cells, proliferative T cells (prol.T), NKT cells, CD8+ T cells, As T cells, helper T cells, naïve T cells, and regulatory T cells (Treg) in Figure 3D, Appendix A. CD8+ T cells expressed several pre-exhausted T cell markers (*ITGAE*, *LAG3*, *GZMH*, etc.), which indicated that CD8+ T cells were pre-exhausted T cells (Appendix A). Additionally, one cluster of T cells expressed several Astrocytes’ markers (*CLU*, *SLC1A3*, *CHI3L1*, etc.), which were termed As T cells (Appendix A). Afterward, cellular composition analysis showed middle and old groups had a gradually higher abundance of CD8+ pre-exhausted T cells (Figure 3E). However, naïve T cells and activated T cells (NKT cells, effector T cells, Tcm, etc.) are enriched in younger populations (Figure 3E). Tumor-associated macrophages (TAMs) further divided into monocyte-derived brain macrophages (Mo-TAMs) expressed monocyte markers (*TGFBI*, *VCAN*, *LYZ*, and *CLEC12A*) and cells termed as Mg-TAMs expressed microglial signature genes (*TMEM119*, *P2RY12*, and *P2RY13*; Appendix A). Cellular composition analysis revealed that brain TAMs derived from monocytes and microglia did not have significant differences (Appendix A). As a result, the younger group had more immune cell types, including naïve cells such as naïve T cells and monocytes (Figure 3E). In contrast, the older group had the fewest activated T cells (Figure 3E). This result is consistent with the previous ssGSEA analysis depicted in Figure 2C. These results suggest that older individuals recruit more exhausted T cells and fewer activated T cells, leading to a decline in immune response to tumor cells.

### 3.4. The Metabolic Reprogramming of Various Cell Types within the TME Exhibits Increased Heterogeneity as the Aging Progresses

Given that GBM malignant cells originate from glial cells, copy number variation (CNV) analysis was performed to distinguish malignant cells from glial cells (Astrocytes, oligodendrocytes, and OPCs) using the ‘copykat’ R package (for details, see the Materials and Methods Section). The CNV level of all glial cells was evaluated and significantly clustered into malignant and non-malignant cells (Appendix A). By choosing top aging-related metabolic switches (adjusted *p*-value < 0.05 based on MMP3C analysis), we found that the metabolic switches of malignant cells were mainly enriched in glucose metabolism and had certain commonalities with each other, such as glycerophospholipid metabolism and oxidative phosphorylation (Figure 3G). Some unique metabolic characteristics are also obvious; for example, OPCs were enriched in Ascorbate and aldarate metabolism (Figure 3G). MMP^3^C analysis showed that malignant OPCs mainly enriched in elder groups exhibited an increased flux of biosynthesis of unsaturated fatty acids and N-glycan in multiple metabolic switches (Figure 3H). In contrast, non-malignant OPCs exhibited the Warburg effect and a decline in the biosynthesis of unsaturated fatty acids and N-glycan (Figure 3H). The results disclosed the metabolic heterogeneity of GBM and indicated that non-malignant cells in GBM in elder populations might intake energy and thus promote tumor progression via the reverse Warburg effect [34].

Next, to investigate which metabolic plasticity is implicated in mainly immune cells in GBM, T cells and macrophages were thereafter selected for further study. Initially, to dissect the evolutionary dynamics of T cell lineages in the aging progression, the pseudo-time cell trajectory analysis of nine cell subclusters was constructed, which generated a three-branch trajectory representing the differentiation process of naïve T cells into either CD8+ pre-exhausted T cells, Tcm cells, or proliferative T cells (Figure 4A). In particular, the two-branch trajectory mapped different aging-related groups (Figure 4A) and evolved the same proliferative T cells with evolution trajectory. Subsequently, we performed an MMP^3^C analysis of each T cell subtype among aged groups. We screened out numerous significant MP pairs through chi-square testing (adjusted *p* value < 0.05; Figure 4B). The activity of glucose metabolic pathways increased, and oxidative phosphorylation reversely decreased in CD8+ pre-exhausted T cells (Figure 4B,C). Furthermore, the correlation analysis assessed the trajectory and the activity score of metabolic switches (Figure 4D). The results showed a metabolic plasticity event composed of Inositol phosphate and oxidative phosphorylation between younger and older groups. Glycolysis and oxidative phosphorylation as the golden standard for tumor metabolism exhibited significance in the T cell’s pseudo-time series, which displayed similar metabolic alterations (Figure 4B,D). The flux of oxidative phosphorylation increased in the young groups but decreased in older groups, which might result in the dysfunction of T cells in elder groups due to the impairment of ATP generation.

Next, we attempted to determine the association between metabolic reprogramming and the function of macrophages within the TME. To this end, we initially conducted another round of cell annotation specifically on macrophages via the mRNA expression of inflammatory factors such as CXCR4, SEPP1, CXCL12, CCL3, CXCL8, CXCL10, and IL1β and anti-inflammatory factors such as CD163, FOLR2, and IL10 (Figure 4E,F and Appendix A). Following this, we plotted a ternary diagram annotated with the percentage of mainly various cell types within each age group, divided by the total number of cells within that group (Figure 4C). Our analysis revealed that the proportion of CCL3+ TAM cells was significantly higher in the young group compared to both the middle-aged and elderly groups (Figure 4C,F). CCL3 is known for its proinflammatory and anticancer properties in GBM [15,31,35,36]. By contrast, FOLR+ TAM, which expressed immune regulatory factors, for example, FOLR, CD163, etc., was reversely enriched in aged groups (Figure 4C,F). Similar to the findings observed in T cells, young patients exhibited superior immune response capabilities, enabling them to react to tumors by supporting proinflammatory processes and maintaining relatively normal energy metabolism. The energy generation pathway (oxidative phosphorylation) significantly increased as opposed to glycolysis, arginine, proline metabolism, etc., in the immune regulatory (FOLR+) TAM in the aged group (Figure 4G). TCA cycle and Terpenoid backbone biosynthesis decreased as opposed to primary bile acid biosynthesis and Glutathione metabolism in CCL3+ TAM in the adulted group (Figure 4G). These metabolic switches, which demonstrate the distinct metabolic adaptations of macrophages, are presented in a way that highlights their potential functional significance.

## 4. Discussion

An increasing body of evidence indicates the significance of Warburg-like metabolic switches in the growth, survival, and treatment of cancer [9,37]. Aging drives novel metabolic reprogramming and gives rise to Warburg-like metabolic switches, which may help create a favorable environment for tumorigenesis and growth. At the same time, the differential changes relative to aging are, to some extent, unclear. Our research, through the MMP^3^C computational framework and rigorous statistical screening, has identified common Warburg-like metabolic switches and key metabolic pathway alterations in aging-related cancers.

Aging can impair oxidative phosphorylation through one of several mechanisms. For instance, with advancing age, there is a gradual decline in phosphoenolpyruvate carboxykinase, accompanied by a corresponding increase in pyruvate kinase, leading to a shift in energy metabolism from oxidative metabolism towards anaerobic glycolysis [38]. Our pan-cancer MP profile of tumors in aged individuals reflects this trend compared to younger individuals, with an increase in glycolysis and a decrease in oxidative phosphorylation (Figure 1E). Also, MMP^3^C analysis of T cells at single-cell resolution supports the impairment of the ATP generation pathway in CD8+ pre-exhausted T cells.

At the single-cell resolution, we observed distinct differentiation processes of T cells under the influence of aging and cancer (Figure 4B,D). It has been known that cancer suppresses glycolysis in activated T cells. Through single-cell sequencing and pseudo-temporal ordering, we found that glycolysis in proliferating T cells was significantly reduced in the middle and old groups compared to the young group. Naturally, the TCA cycle also exhibited a substantial decline. This led to a general decrease in biosynthesis and metabolism of proliferating T cells in both the middle and old groups (Figure 4B). In addition, the immune-activated and regulatory TAMs also exhibited similar distribution among aging-related groups.

Indeed, there are certain limitations to our research. First and foremost, while our methodology effectively captures metabolic plasticity events with statistical significance, it falls short of precisely dissecting the underlying causes of these events, necessitating additional molecular experimental validation. Secondly, the aging-specific conditions under which our single-cell sequencing was conducted resulted in a relatively small sample size of T cells and their diverse subpopulations, potentially impacting the precision of our findings. However, this discovery could still enhance our understanding of the complex interplay between aging and cancer metabolism and unveil new avenues for the development of targeted therapeutic strategies.

## 5. Conclusions

In this study, we investigated the association between metabo-plastic pairs derived from 84 KEGG MPs and aging based on MMP^3^C analysis using bulk RNA-seq. The results exhibited increased fluxes of glucose-related pathways compared with ATP generation pathways. Meanwhile, single-cell transcriptome profiling with approximately 180,000 individual cells was utilized to characterize the aging-induced metabolic switches in the TME. This revealed that the metabolic dysfunction of immune cells (T cells and macrophages) might lead to TME reprogramming and thus provide a perfect environment for malignant cells in elder individuals. In summary, with the increase in life expectancy, understanding the interaction between aging-related metabolic reprogramming and immune cells will aid in the development of precise treatment strategies for elderly patients.

## Figures and Tables

**Figure 1 cells-13-01721-f001:**
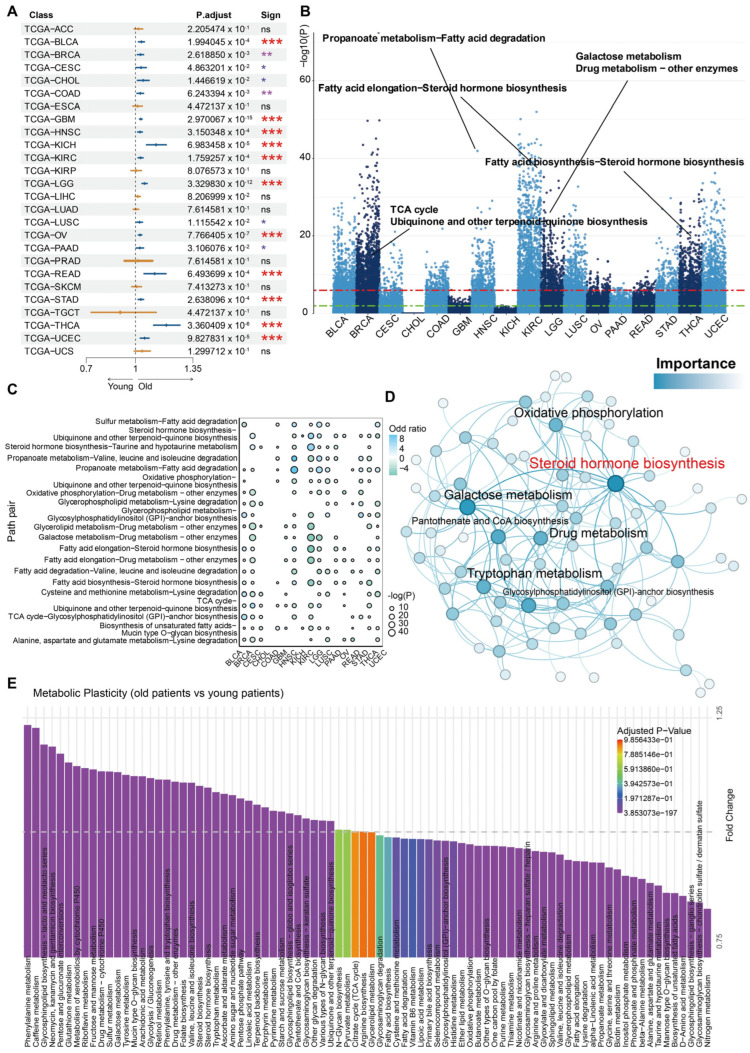
Identifying the correlation between tumor heterogeneity and aging and characterizing the dynamic metabolic plasticity of distinct metabolic pathways via pan-cancer bulk sequencing. (**A**) Forest plots were used to identify tumor types associated with advanced aging. (**B**) A Manhattan plot illustrates the disparities in potential metabolic plasticity events among 17 distinct cancer types within the pan-cancer cohort. (**C**) Visualizing common metabolic plasticity events across multiple cancer types through dot plots. (**D**) Crucial pathway nodes within the altered metabo-plastic architecture of tumors have been identified through network analysis. (**E**) Relative metabolic pathway activity scoring reveals metabolic plasticity. The labeled asterisk indicated the statistical *p*-values (ns *p* > 0.05, * *p* < 0.05, ** *p* < 0.01, and *** *p* < 0.001).

**Figure 2 cells-13-01721-f002:**
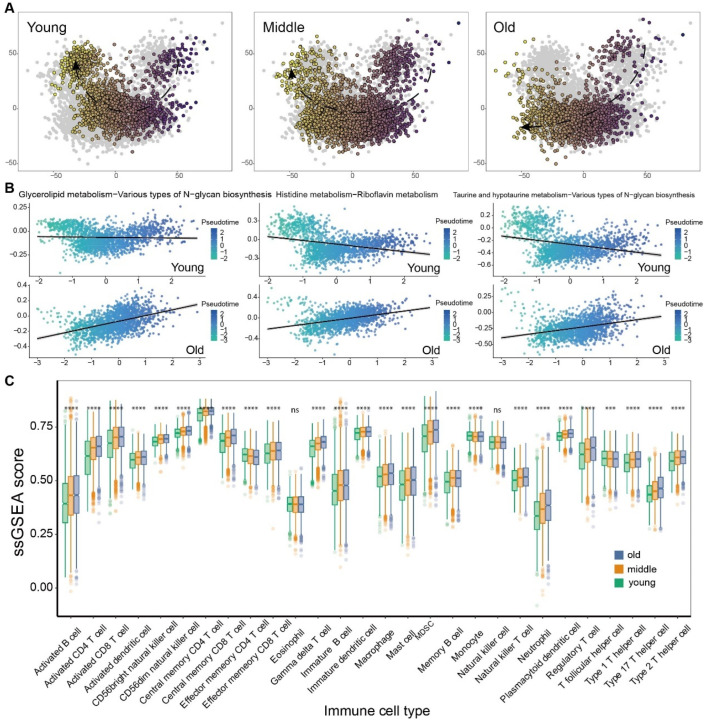
Large-sample bulk RNA-seq pseudo-time analysis and ssGSEA immune infiltration analysis revealed unique metabolic characteristics in different aging-related subgroups. (**A**) Pseudo-temporal analysis across three distinct aging-related groups elucidated distinct metabolic trajectories. (**B**) PhenoPath-guided pseudo-temporal analysis identified metabolic transitions associated with temporal variations. (**C**) ssGSEA analysis revealed the immune infiltration scores across diverse cell types. The labeled asterisk indicated the statistical *p*-values (ns *p* > 0.05, *** *p* < 0.001 and **** *p* < 0.0001).

**Figure 3 cells-13-01721-f003:**
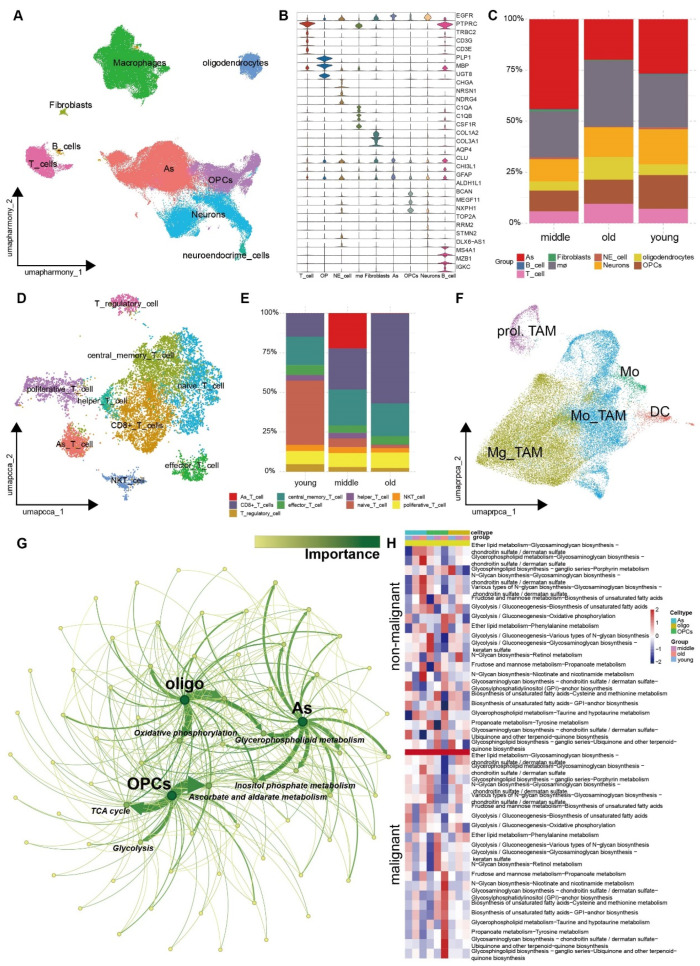
The identification of the landscape of various cell types within the TME among aging-related groups at the single-cell resolution. (**A**) UMAP plot of 181,286 cells from 24 primary glioma patients displaying nine main cell types. (**B**) Violin plot unveiling different kinds of cell types in the TME. (**C**) Percent bar plot unveiling the distribution of main cell types in the TME. (**D**) UMAP plot of 12,144 T cells displaying nine main T cell types. (**E**) Percent bar plot unveiling the distribution of T cells among aging-related groups. (**F**) UMAP plot of 45,074 TAMs displaying five main cell types. (**G**) Network analysis revealed crucial pathway nodes in aging-related altered metabolic plasticity architecture. (**H**) Heatmap unveiling unique metabolic plasticity events in non-malignant and malignant cell types among aging-related subgroups.

**Figure 4 cells-13-01721-f004:**
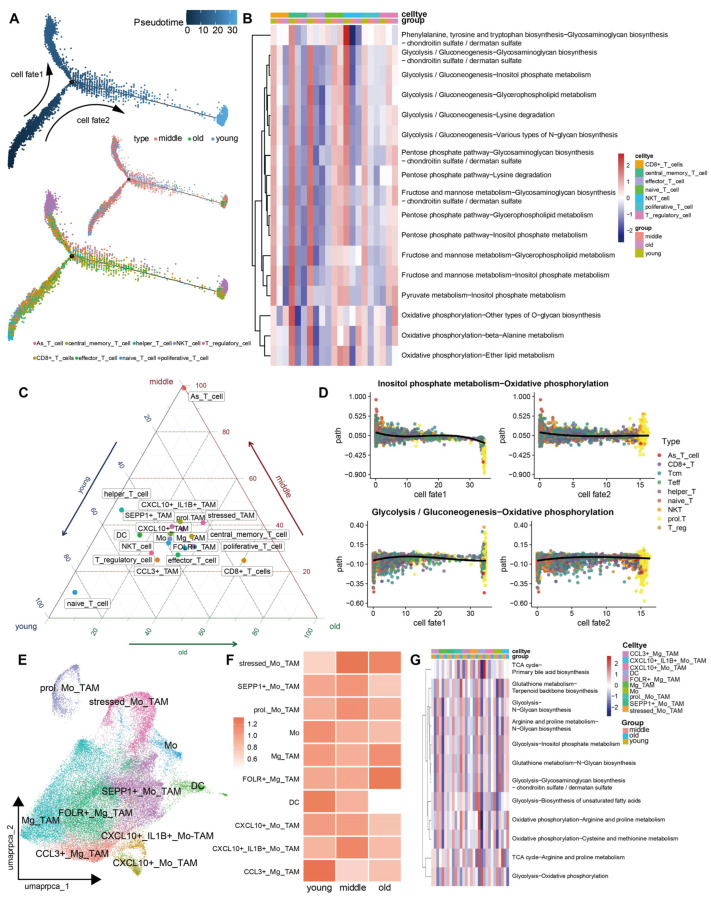
The identification of metabolic heterogeneity within various immune cell types among aging-related groups. (**A**) The potential trajectory of all T cells identified three distinct cell fates. (**B**) Heatmap unveiling specific metabolic plasticity events in multiple T cell subtypes among aging-related subgroups. (**C**) Aging-related subgroups enrichment of TAM types and T cell types. (**D**) Dot plots of dynamic activity difference in metabolic plasticity events along two cell fates consistent with aging-related groups. (**E**) UMAP plot of 45,074 TAMs displaying various functional cell types. (**F**) Heatmap unveiling the distribution of different functional TAMs among aging-related groups. (**G**) Heatmap unveiling specific metabolic plasticity events in TAMs with different immune functions among aging-related subgroups.

## Data Availability

All real datasets used within this study are retrievable from public databases with details provided within the Appendix A.

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
