# Peer review of "Metabolic Reprogramming Induced by Aging Modifies the Tumor Microenvironment"

_cells, 2024, doi:10.3390/cells13201721_

Round 1
Reviewer 1 Report
Comments and Suggestions for Authors
- Introduction should be without the text about methods and results.
- Please define the aim of the study clearly.
- Please check the manuscript again, instead of the number of reference there is "ref" in brackets.
Comments on the Quality of English Language- English should be improved.
Reviewer 2 Report
Comments and Suggestions for Authors
Aging plays a significant role in cancer progression, with metabolic reprogramming being a key factor. The authors utilized various computational approaches to analyze age-related metabolic alterations across 17 cancer types in older patients. They found that certain pathways, such as glycolysis and oxidative phosphorylation, were upregulated. Notably, some immune cells also exhibited changes in signaling pathways, contributing to the remodeling of the tumor microenvironment (TME). Targeting metabolism in the TME could offer promising avenues for cancer therapy.
Comments on this manuscript: The study primarily relies on RNA sequencing data; incorporating proteomics databases could strengthen the conclusions. Additionally, using other validation datasets would provide further support for the findings. Why are these ages 50 and 65 are used for the "young," "middle-aged," and "old" .Please review the manuscript for grammatical errors and typos and consider increasing the resolution of the figures.
Comments on the Quality of English LanguageNONE
